# Anticancer Activity of Delta-Tocotrienol in Human Hepatocarcinoma: Involvement of Autophagy Induction

**DOI:** 10.3390/cancers16152654

**Published:** 2024-07-26

**Authors:** Marina Montagnani Marelli, Chiara Macchi, Massimiliano Ruscica, Patrizia Sartori, Roberta Manuela Moretti

**Affiliations:** 1Department of Pharmacological and Biomolecular Sciences “Rodolfo Paoletti”, Università degli Studi di Milano, 20133 Milan, Italy; chiara.macchi@unimi.it (C.M.); massimiliano.ruscica@unimi.it (M.R.); roberta.moretti@unimi.it (R.M.M.); 2Department of Cardio-Thoracic-Vascular Diseases-Foundation, IRCCS Cà Granda Ospedale Maggiore Policlinico, 20162 Milan, Italy; 3Department of Biomedical Sciences for Health, Università degli Studi di Milano, 20133 Milan, Italy; patrizia.sartori@unimi.it

**Keywords:** hepatocarcinoma, delta-tocotrienol, apoptosis, autophagy, mitochondria

## Abstract

**Simple Summary:**

Hepatocellular carcinoma (HCC) is the predominant form of primary liver cancer (about 85–90%). In the advanced stage of the disease, existing therapies cause toxic side effects and patients often develop chemoresistance. It is therefore important to identify new compounds with low toxicity that can be used in patients with compromised liver and advanced HCC. The objective of this research was to study the antitumoral activity of delta-tocotrienol, a natural compound derived from Vitamin E, on human hepatocarcinoma cell lines. This study supports the evidence that this compound exerts an antitumoral action activating the autophagic process, leading to cancer cell death. We believe that these data may provide a basis for considering delta-tocotrienol as a potential adjuvant therapy for the treatment of advanced HCC.

**Abstract:**

(1) Hepatocellular carcinoma (HCC) is the predominant form of primary liver cancer. Surgical resection, tumor ablation, and liver transplantation are curative treatments indicated for early-stage HCC. The management of intermediate and advanced stages of pathology is based on the use of systemic therapies which often show important side effects. Vitamin E-derivative tocotrienols (TTs) play antitumoral properties in different tumors. Here, we analyzed the activity of delta-TT (δ-TT) on HCC human cell lines. (2) We analyzed the ability of δ-TT to trigger apoptosis, to induce oxidative stress, autophagy, and mitophagy in HepG2 cell line. We evaluated the correlation between the activation of autophagy with the ability of δ-TT to induce cell death. (3) The data obtained demonstrate that δ-TT exerts an antiproliferative and proapoptotic effect in HCC cells. Furthermore, δ-TT induces the release of mitochondrial ROS and causes a structural and functional alteration of the mitochondria compatible with a fission process. Finally, δ-TT triggers selective autophagy process removing dysfunctional mitochondria. Inhibition of autophagy reversed the cytotoxic action of δ-TT. (4) Our results demonstrate that δ-TT through the activation of autophagy could represent a potential new approach in the treatment of advanced HCC.

## 1. Introduction

Hepatocellular carcinoma (HCC) is the predominant form of primary liver cancer (about 85–90%). The risk factors that most influence the development of HCC are hepatitis B virus (HBV) or hepatitis C virus (HBC) infection, alcohol consumption, metabolic dysfunction-associated fatty liver disease (MAFLD) [1], and nonalcoholic steatohepatitis (NADH). In addition, the coincident comorbidity with obesity and diabetes mellitus predisposes to progression of cirrhosis and consequently to HCC [2,3]. Barcelona Clinic Liver Cancer (BCLC) staging is the most used system to classify HCC and to decide the best therapeutic strategy at different stages of disease [4]. Surgical resection, tumor ablation, and liver transplantation are curative treatments indicated for very early and early-stage HCC. Unfortunately, a significant proportion of patients present intermediate and advanced stage of disease and curative treatments are frequently precluded [5]. The only approved systemic treatments for advanced HCC include targeted therapeutic agents, e.g., multikinase inhibitors Sorafenib and Lenvatinib or immune checkpoint inhibitor Atezolizumab (anti-PD1) in combination with Bevacizumab (VEGF inhibitor) for first-line therapies. At present, the second-line therapies approved for the treatment of advanced HCC after Sorafenib are Regorafenib and Cabozantinib (multi-kinase inhibitors), Nivolumab (anti-PD1), Pembrolizumab (anti-PD1), Ramucirumab (VEGFR-inhibitor), and Ipilimumab (anti CTLA4) in combination with Nivolumab [6]. While these agents prolong patients’ survival, they are also associated with serious side effects and limited efficacy due to the development of chemoresistance. For this reason, it is mandatory to unveil new and more tolerated therapeutic approaches in the advanced stages of HCC. Within this framework, the autophagy represents an important biological process involved in tumor development and progression, as well as in response to pharmacological therapies [7].

Macroautophagy (hereafter called autophagy) is a conserved non-selective mechanism where misfolded macromolecules and dysfunctional organelles undergo lysosomal degradation; different forms of autophagy are selective for different intracellular organelles such as mitochondria (mitophagy), peroxisomes (perophagy), or lipids droplets (lipophagy) [8,9].

Autophagy, which is a complex process that plays a dual role in cancer progression, is strictly dependent on cancer type, stage, and genetics. A wide variety of studies elucidated the effects of autophagy on HCC, that could act as a tumor suppressor factor, especially in the early stages of tumorigenesis by limiting inflammation, removing dysfunctional cellular components, reducing reactive oxygen species (ROS), and genomic damage during stress, thus preventing tumor initiation [10,11]. On the other hand, at late stages of hepatocarcinogenesis, autophagy could support malignant cells transformation by facilitating their survival by counteracting cell death [8,12].

Autophagy and apoptosis are two processes that regulate cell survival or cell death by controlling cellular and tissue homeostasis [13]. These molecular pathways may be independent or interconnected; in fact, autophagy can both inhibit or promote apoptosis in cancer cells. Whether autophagy is associated with cell death during cancer treatment or with cell survival and resistance to anticancer treatments is still a matter of debate [7,14]. Indeed, numerous studies, focusing on HCC, have highlighted that autophagy correlates with chemoresistance, including resistance to Sorafenib, and that autophagy inhibitors can sensitize cancer cells to anticancer treatments [15,16]. Conversely, in HCC, some substances with antitumor activity induce apoptotic cell death in concert with autophagy [13]. Many natural compounds possess antitumoral activity by activating both apoptosis and autophagy and among these are Vitamin E derivatives.

Vitamin E is composed of tocopherols (TPs) and tocotrienols (TTs), each of which presents four isomers: α, β, γ, and δ [17]. In particular, numerous research showed that both δ-TT and γ-TT exert a marked anticancer activity in different types of in vitro and in vivo models of tumors [18] through cytotoxic mechanisms that involve various types of cell death, both conventional (apoptosis) [19,20] and non-conventional (paraptosis and necroptosis) [21,22,23]. Furthermore, previous studies have also highlighted that TTs are able to induce cell death by modulating both endoplasmic reticulum (ER) stress and autophagy [22,24,25,26]. In HCC δ-TT and γ-TT exert antitumoral, antiangiogenic and proapoptotic activity as well as reduce the intracellular level of triglycerides and cholesterol [27,28,29,30,31,32].

Then, with the aim of analyzing the antitumoral action of δ-TT in human hepatocarcinoma cells, we demonstrated that δ-TT exerts its action by altering the mitochondrial structure and dynamism, increasing ROS release. Furthermore, we highlighted that the cytotoxic action of δ-TT is related to autophagy activation, which determines a massive removal of dysfunctional mitochondria. Our results convey the idea that δ-TT could be a promising therapeutic approach in advanced hepatocarcinoma.

## 2. Materials and Methods

### 2.1. Antibodies and Reagents

The δ-TT was purified from a commercial extract of Annatto (*Bixa orellana* L.) seeds (American River Nutrition Inc., Hadley, MA, USA) as previously described by our group [33]. The primary antibodies anti-cleaved-caspase-3 (Asp-175) (#9664), anti-cleaved-PARP (#5625), anti-OPA1 (D6U6N) (#80471), anti-Mitofusin-2 (MNF2) (D2D10) (#9482), anti-Parkin (Prk8) (#4211), anti-BNIP3 (D7U1T) (#44060), anti-Atg5 (#12964), rabbit and mouse horseradish-peroxidase-conjugated secondary antibodies were from Cell Signaling Technology Inc. (Boston, MA, USA). The primary antibodies anti-cytochrome *c* (sc-13560), anti-procaspase-3 (E-8) (sc-7272), anti-Fis1 (B-5) (sc-376447), anti-DRP1 (C-5) (sc-271583), and anti-TOM20 (F-10) (sc-17764) were from Santa Cruz Biotechnology Inc. (Santa Cruz, CA, USA). Anti-LC3 (L8918) and anti-alpha-tubulin (T6199) were purchased from Sigma-Aldrich (St. Louis, MO, USA). Anti-SQSTM1/p62 (PA5-20839) was from ThermoFisher Scientific (Waltham, MA, USA). FITC-conjugated secondary antibody Alexa Fluor 488, MitoTracker Orange CMTM Ros (mitochondrial selective dye) (M7510) and the LysoTracker Green DND-26 (lysosomal selective dye) (L7526) were acquired from Molecular Probes (Thermo-Fisher Scientific, Waltham, MA, USA). 3-Methyladenine (3-MA) (S2767) was from Selleckchem (Munich, Germany). 3-(4,5)-dimethylthiazol-2-yl-2,5-diphenyltetrazolium bromide (MTT), Trypan blue (T8184), chloroquine (CQ) (C6628), and N-acetyl-L-cysteine (NAC) (A 8199) were purchased from Sigma-Aldrich (St. Louis, MO, USA).

### 2.2. Cell Culture

Human HepG2 cells was purchased from the American Type Culture Collection (ATCC, Manassas, VA, USA). HepG2 cells were grown in Dulbecco’s Modified Eagle’s Medium (DMEM) supplemented with sodium pyruvate, 20% fetal bovine serum (FBS), glutamine (1 mmol/L), and antibiotics (100 IU/mL penicillin G sodium and 100 μg/mL streptomycin sulfate). Human Huh7 cells were donated from Prof. Nicola Ferri (Università degli Studi di Padova, Padua, Italy). Huh7 cells were grown in Dulbecco’s Modified Eagle’s Medium Low Glucose supplemented with sodium pyruvate, 10% FBS, glutamine (1 mmol/L), and antibiotics (100 IU/mL penicillin G sodium and 100 μg/mL streptomycin sulfate). Cells were cultured in humidified atmosphere of 5% CO_2_/95% air at 37 °C.

### 2.3. Proliferation Assays

In order to assess the effect of δ-TT on cell proliferation, HepG2 and Huh7 cells were seeded in 6 cm dishes (15 × 10^4^ cells/ dish and 8 × 10^4^ cells/dish, respectively). Cells were then treated with δ-TT (10, 15, 20, 25 μg/mL) or with vehicle (DMSO) for 48 h. At the end of the treatments, cells were harvested by trypsinization and counted by hemocytometer in presence of Trypan blue stain 0.4%. Cell count results shown are indicative of viable cells only.

### 2.4. MTT Viability Assay

HepG2 and Huh7 cells were seeded in 24-well culture plates at a concentration of 3 × 10^4^ cells/well. Cells were treated with δ-TT (10, 15, 20, 25 μg/mL) for 48 h. After treatment, cells were incubated in MTT solution (0.5 mg/mL) dissolved in medium without phenol red and FBS at 37 °C for 30 min. Later, culture media were removed and changed with isopropanol to dissolve the crystals. The OD values were measured at 550 nm with EnSpire Multimode Plate reader (PerkinElmer, Milano, Italy).

### 2.5. Clonogenic Assay

HepG2 cells were seeded in 6-well plates at a density of 500 cells/well. After 72 h of the treatment with δ-TT (20 and 25 μg/mL), the medium was removed, and colonies formation was performed for 11 or 18 days. Clones were fixed with 70% methanol and stained with Crystal Violet 0.15%. A Nikon (Tokyo, Japan) photo camera was used to capture images of stained clones.

### 2.6. Western Blot

For Western blot (WB) experiments, HepG2 cells were seeded in 6 cm dishes at the density of 8 × 10^4^ cells/dish. At the end of each treatment, adherent and floating cells were harvested and lysed in RIPA buffer (0.05 mol/L Tris HCl pH 7.7, 0.15 mol/L NaCl, 0.8% SDS, 10 mmol/L EDTA, 100 μmol/L NaVO_4_, 50 mmol/L NaF, 0.3 mmol/L PMSF, 5 mmol/L iodoacetic acid) containing leupeptin (50 μg/mL,), aprotinin (5 μL/mL), and pepstatin (50 μg/mL). Protein extract concentration was determined using BCA protein assay kit (Euroclone). Protein extracts (20–35 μg) were resolved on SDS gel electrophoresis and transferred to nitrocellulose or PVDF (for LC3 analysis) membranes. After blocking, membranes were incubated overnight at 4 °C using the specific primary antibodies. Anti-rabbit or anti-mouse HRP secondary antibodies were incubated at room temperature for 1 h. Chemiluminescence analysis was evaluated using the kit Westar ETAC Ultra 2.0 (XLS0750100) (Cyanagen, Bologna, Italy). Tubulin-alpha expression was evaluated as a loading control in all WB experiments. Relative optical density of the bands was determined by ImageJ software (ImageJ 1.50i). Uncropped WB images are available in Appendix A.

### 2.7. Flow Cytometry Analysis (FACS)

HepG2 cells were seeded in 6 cm dishes at a density of 1.5 × 10^5^ cells/plate and treated for 72 h with δ-TT (20–25 μg/mL). Successively, the cells were collected and incubated with Annexin/PI according to the manufacturer’s instructions of AnnexinV-FITC/PI apoptosis detection kit (eBioscience, San Diego, CA, USA). The stained cells were analyzed by flow cytometry NovoCyte Flow Cytometer 3000 (Acea Bioscience, Inc., San Diego, CA, USA) and results were analyzed by software Novo Express (Version 1.4.1).

### 2.8. Phase-Contrast Microscopy

Cell morphology was analyzed by phase-contrast microscopy. HepG2 cells were seeded at 10 × 10^4^ cells/well in 6-well plates and treated with δ-TT (20 and 25 μg/mL) for 48 h. Morphological analysis was performed by optical microscopy from different fields under a Zeiss Axiovert 200 microscope (Zeiss, Oberkochen, Germany) with a 20 × 0.4 objective lens linked to a CoolSnap Es CCD camera (Roper Scientific-Crisel Instruments, Rome, Italy).

### 2.9. Transmission Electron Microscopy (TEM)

For TEM analysis, HepG2 cells were seeded in 6 plates at 3 × 10^5^ cells/ plate and treated with δ-TT (25 μg/mL) for 24 h and 48 h. Cell pellets were fixed overnight in a solution containing 2.5% glutaraldehyde in 0.1 M sodium cacodylate buffer (pH 7.3). Samples were then washed in cacodylate buffer and fixed in 1% osmium tetroxide (Sigma-Aldrich) in the same buffer at 0 °C for 90 min. After rinsing in distilled water pellets were stained in 2% aqueous uranyl acetate, dehydrated in a graded acetone series, and embedded in Epon-Araldite resin. Ultrathin sections were obtained with a Leica Supernova ultramicrotome (Reichert Ultracut, Wien, Austria) and counterstained with lead citrate. TEM investigations were performed with a Zeiss EM10 electron microscope (Carl Zeiss, Oberkochen, Germany).

### 2.10. Mitochondrial Staining

HepG2 cells were plated in 24-multiwell at 3 × 10^4^ cells/well on coated polylysine coverslips. After the treatment with δ-TT (25 μg/mL, 24 h and 48 h), cells were incubated with MitoTracker Orange (100 nM) for 30 min. Then, the cells were fixed with paraformaldehyde and analyzed using Zeiss Axiovert 200 microscope (Zeiss, Oberkochen, Germany) with 63×/1.4 objective lens linked to a Coolsnap Es CCD camera (Ropper Scientific-Trenton, NJ, USA).

### 2.11. Mitochondrial Superoxide Analysis

To detect mitochondrial superoxide, HepG2 cells were seeded at 15 × 10^4^ cells/well in 6-well plates. After 24 h of treatment with δ-TT (25 μg/mL), cells were stained with 1.25 μM MitoSOX Red indicator (Thermo Fisher Scientific) for 30 min at 37 °C. Cells were centrifuged and washed with PBS. MitoSOX fluorescence was analyzed by flow cytometry (NovoCyte Flow Cytometer 3000), by measuring the fluorescence absorption and emission at 396 nm and 610 nm, respectively.

### 2.12. TMRE-Mitochondrial Membrane Potential Assay

To analyze mitochondrial membrane depolarization, the MitoPT TMRE Assay (ImmunoChemistry Technologies, Davis, CA, USA) was used. HepG2 cells were seeded at 15 × 10^4^ cells/well in 6-well plates and treated with δ-TT (25 μg/mL, 48 h). As a positive control of reduced mitochondrial potential, HepG2 were exposed to CCCP (carbonyl cyanide 3-chlorophenylhydrazone) 50 μM for 30 min at 37 °C. Cells were stained with the fluorescent dye TMRE 200 nM for 20 min at 37 °C. Cells were then collected, centrifuged, washed, and the fluorescence (549 nm excitation and 574 nm emission) was analyzed by flow cytometry (NovoCyte Flow Cytometer 3000).

### 2.13. Immunofluorescence Analysis

For immunofluorescence studies, HepG2 cells were seeded in 24-well plates at 3 × 10^4^ cells/well on polylysine-coated coverslips. After, the treatments with δ-TT cells were fixed (3% paraformaldehyde/2% sucrose). Cells were then permeabilized (0.1% Triton X-100 in PBS) for 20 min and incubated in blocking solution (1% horse serum in PBS) for 1 h. Primary antibodies diluted in PBS with 3% BSA were added overnight at 4 °C. The cells were washed with PBS and incubated with FITC-conjugated secondary antibody Alexa Fluor (Molecular Probes-Thermo-Fisher Scientific) for 1 h at room temperature. Nuclei were stained with Hoechst. Zeiss Axiovert 200 microscope with 63×/1.4 objective lens linked to a Coolsnap Es CCD camera was used to analyze cell fluorescence.

### 2.14. Small Interfering RNA (siRNA)

To silence endogenous ATG5 expression, negative control (NC) *siRNA* (#6568) and *ATG5 siRNA I* (#6345) obtained from Cell Signaling were used. HepG2 were seeded in 6-well culture plates at a concentration of 8 × 10^4^ cells/well and transfected with NC (50 nM) or *ATG5 siRNA* (50 nM) by Lipofectamine 3000 (ThermoFischer Scientific) according to the manufacturer’s instructions. Medium was replaced with treatments after 24 h.

### 2.15. Mitochondrial and Lysosomal Staining

HepG2 cells were plated in 24-well plates at 3 × 10^4^ cells/well on coated polilysine coverslips. After the treatment with δ-TT (25 μg/mL, 48 h), the cells were incubated with MitoTracker Orange as described above and with 100 nM LysoTracker Green for 45 min to stain the lysosomes. Successively, the cells were fixed with paraformaldehyde and analyzed using Zeiss Axiovert 200 microscope with 63×/1.4 objective lens linked to a Coolsnap Es CCD camera.

### 2.16. Statistical Analysis

All experiments were performed three times and the results were analyzed by one-way analysis of variance (ANOVA) followed by Dunnett’s test or Bonferroni test. When two groups were present, data were analyzed by Mann–Whitney U test for non-parametric distribution. Prism software was used for the analyses (Prism 8 for Mac OS version 8.2.1, GraphPad Software San Diego, CA, USA).

## 3. Results

### 3.1. δ-TT Induces Antitumoral Effect in HCC Cells

The HepG2 and Huh7 cells were treated with δ-TT (10–25 μg/mL) to evaluate its effect on cell growth and viability. Treatment for 48 h revelated that δ-TT significantly inhibited both cell proliferation (Figure 1A) and viability (Figure 1B) in a dose-dependent (range 15–25 μg/mL) manner in both cell lines.

Subsequent experiments were conducted on the HepG2 cell line. In this cell line, preliminary experiments were conducted to analyze the effect of δ-TT (25 μg/mL) at different times (24 h, 48 h and 72 h). The results of cellular growth are shown in Appendix A.

To analyze the cytotoxic action of δ-TT, clonogenic assays were conducted on the HepG2 cell line treating the cells for 72 h (doses of 20 and 25 μg/mL). At the end of the treatment, the culture medium was removed, and the cells were left to grow for 11 or 18 days and finally fixed and stained with Crystal Violet. The analysis of colonies highlighted that treatment with δ-TT, at both 20 and 25 μg/mL significantly reduced the number of colonies, confirming ability of δ-TT to exert a cytotoxic effect on HCC cells (Figure 1C).

We therefore analyzed the molecular mechanism involved in cytotoxic activity of δ-TT. Our attention was focused on a possible apoptosis activation. FACS analysis by Annexin-FITC/Propidium Iodide (PI) double staining showed that 72 h treatment with δ-TT, at both 20 and 25 μg/mL, induced an accumulation of cells in early and late apoptosis, compared to control cells (Figure 1D). The execution of apoptosis is triggered by activation of caspase-3, so we analyzed this protein in its inactive (procaspase-3) and active (cleaved caspase-3) form by WB. As shown in Figure 1E, caspase-3 was cleaved after δ-TT treatment with both 20 and 25 μg/mL for 72 h. Then, to assess whether the intrinsic mitochondrial pathway of apoptosis was involved, we evaluated the release of cytochrome *c* from mitochondria. Figure 1F showed that the treatment with δ-TT (20 and 25 µg/mL) for 48 h modified the co-localization of cytochrome *c* (green) with MitoTracker stained mitochondria (red) compared to control cells. In particular, in controls, cytochrome *c* was co-localized with mitochondria, giving rise to a merge showed by a yellow color. Treatment induced a loss of co-localization demonstrating that cytochrome *c* was released from the mitochondrial inner membrane. Furthermore, MitoTracker staining highlighted a marked modification of the mitochondrial morphology. Overall, these results demonstrate that the cytotoxic action of δ-TT is due to the activation of the intrinsic apoptosis pathway. The obtained results are in line with previously studies reporting that γ-TT suppressed HCC cell growth by the activation of apoptosis [29,31].

### 3.2. δ-TT Induces Mitochondrial Phenotypic Alteration in HCC Cells

We analyzed by phase-contrast microscopy the cellular phenotypic morphology of HepG2 cells treated with δ-TT (20 and 25 μg/mL) for 48 h, and we observed significant cytoplasmic vacuolization in addition to vacuoles physiologically present in control cells (Figure 2A). To better clarify the nature of this phenotypic modification, we also conducted TEM analysis. After treatment with δ-TT (25 μg/mL), we found cytoplasmic alteration and dilated mitochondria with different degrees of dysfunctionality in the mitochondrial cristae (“swollen” or “vesicular” mitochondria) (Figure 2B).

### 3.3. δ-TT Induces Mitochondrial Fission and ROS Release in HCC Cells

To better investigate the mitochondrial morphology, we stained the HepG2 cells with MitoTracker after treatment with δ-TT (25 μg/mL) for 24 h and 48 h. The immunofluorescence analysis showed that, at both times, the control cells presented cytoplasmatic and uniform distribution of mitochondria, with the typical elongated conformation and a well-defined concatenation. Conversely, in treated cells, important changes in mitochondrial morphology were observed. These organelles had a fragmented distribution and rounded conformation with different degrees of dilatation. This evidence supports the previous one in which the treatment with δ-TT changed mitochondrial architecture (Figure 3A). Mitochondria are subjected to continuous dynamism, which involves fission and fusion processes [34]; when these organelles are dysfunctional, they modify the balance of this dynamism, moving more towards the phenomenon of mitochondrial fission rather than fusion. For this reason, we analyzed the expression of the main proteins involved in this phenomenon. The HepG2 cells were treated with δ-TT at the dose of 25 μg/mL for 24 h and 48 h, and the expression levels of fission proteins compared to fusion proteins were evaluated. The result of this analysis, presented in Figure 3B, shows that, at both times considered, the treatment induced a significant increase in expression of both the Fission 1 homolog protein (Fis1) and Dynamin-related protein 1 (DRP1), suggesting an increase in the phenomenon of mitochondrial fission. At the same time, there was a decrease in the expression of optic atrophy protein 1 (OPA1) and mitofusin 2 (MFN2), proteins involved in mitochondrial fusion. The expression of all these proteins was compared with TOM20, a constitutive protein of the mitochondrial outer membrane, and with tubulin, a constitutive cellular protein. Overall, the results obtained suggest that treatment with δ-TT increases mitochondrial fission.

In order to evaluate the process involved in these mitochondrial phenotypic changes, we analyzed HepG2 cells previously treated with δ-TT (25 μg/mL for 24 h) in association with NAC (4 mM), an antioxidant substance. The MitoTracker staining reported in Figure 3C, shows that the treatment with NAC reverted the δ-TT-induced morphological changes in mitochondria. This result leads us to hypothesize that δ-TT could generate an increase in cellular oxidative stress due to an overproduction of ROS and a simultaneous inefficiency of the antioxidant systems. We therefore evaluated the release of mitochondrial ROS after treatment with δ-TT 25 μg/mL for 24 h by flow cytometry using MitoSOX assay. Figure 3D shows that δ-TT increased mitochondrial ROS release compared to control cells and this was associated with an alteration of mitochondrial membrane potential (ΔΨm). Indeed, treatment with δ-TT 25 μg/mL for 48 h significantly reduced ΔΨm assessed as TMRE assay (Figure 3E), confirming the alteration of mitochondrial functionality. All-in-all, these results show that δ-TT treatment generated dysfunctional and damaged mitochondria, leading to ROS and cytochrome *c* release and consequently, the induction of cellular apoptosis [35].

### 3.4. δ-TT Induces Autophagy in HCC Cells

Autophagy is a complex phenomenon that allows cells to maintain the correct turnover of cellular structures or organelles. Within the portrait of cellular homeostasis, autophagy is a crucial response to oxidative stress related to an increase in ROS [36].

To evaluate a possible autophagy activation, we treated the HepG2 cells with δ-TT at a dose of 25 μg/mL for 24 h and 48 h and subsequently analyzed the expression and distribution of microtubule-associated protein1A/1B-light chain 3 (LC3) and the sequestosome-1 (SQSTM1/p62). Following autophagic activation, LC3 is converted from its diffuse cytoplasmic LC3-I form to LC3-II, which represents the lipidated form integrated into the nascent autophagosomes. Figure 4A shows that the expression of LC3-II increased more than LC3-I after δ-TT treatment, both at 24 h and 48 h, resulting in the significant increase in the LC3-II/LC3-I ratio; this result attests that δ-TT induces autophagosomes formation. Next, we analyzed the autophagy adaptor p62, a protein which recognizes and binds the cellular material to insert into autophagosomes. WB analysis showed that p62 expression was upregulated after treatment with δ-TT (25 μg/mL, 24 h and 48 h) (Figure 4B). Moreover, immunofluorescence analysis after treatment with δ-TT (25 μg/mL, 48 h) confirmed the presence of LC3 puncta in treated cells compared to control cells, where LC3 was diffuse and homogeneous in the cytoplasm (Figure 4C) and showed “p62 bodies”, representative of p62 accumulation into autophagosome in treated cells (25 μg/mL, 48 h) (Figure 4D). Furthermore, to exclude that the increase in LC3 and p62 expression was caused by an impairment of autophagic flux, we evaluated the expression of LC3 and p62 after co-treatment with δ-TT (25 μg/mL, 48 h) and CQ (10 μM, last 24 h) (inhibitor of lysosomal enzymes activity). The increased expression of both autophagy markers in presence of δ-TT and CQ co-treatment compared to δ-TT alone demonstrated that autophagic flux was active and it determined the effective degradation of materials accumulated into autophagosomes at the lysosomal level (Figure 4E). Overall, we can postulate that δ-TT induces autophagy in HCC cells.

### 3.5. Effect of Inhibition of Autophagy δ-TT-Induced on Survival of HCC Cells

In cancer cells, autophagy carries out a complex and dual role; in fact, it can develop a cytoprotective action and drug resistance, and on the other hand, in different cellular conditions, cell death can be induced [37].

We then analyzed the cytotoxic activity of δ-TT in HepG2 cells in the presence of 3-MA, an autophagy inhibitor which blocks the formation of autophagosomes. Preliminarily, we verified whether 3-MA at a dose of 2 mM was able to block autophagy activation. WB analysis of LC3 in Figure 5A showed that co-treatment with 3-MA and δ-TT impaired the LC3-II accumulation. We then performed cell proliferation analysis in cells treated with δ-TT (25 µg/mL, 48 h) in the presence of 3-MA. The results obtained showed that 3-MA reversed, in a significative manner, the antiproliferative (Figure 5B) and proapoptotic activity (Figure 5C) of δ-TT. These results demonstrated that δ-TT-induced autophagy is a process that triggers apoptotic cell death. Since 3-MA could also inhibit other cellular functions, we genetically inhibited autophagy by knockout of *ATG5* gene using *ATG5 siRNA*. The silencing of ATG5 protein expression was assessed by WB after 72 h of silencing using *ATG5 siRNA* (Figure 5D). The result in Figure 5E confirmed that LC3-II formation induced by δ-TT (25 µg/mL, 48 h) was significant reduced after *ATG5 siRNA* (Figure 5E). In addition to this, the antiproliferative effect and proapoptotic activity of δ-TT (25 µg/mL, 48 h) (Figure 5F,G) in the presence of *ATG5 siRNA* were counteracted, similar to the result obtained with 3-MA. These results collectively demonstrate that δ-TT-induced autophagy is a process that promotes apoptotic cell death in HepG2.

### 3.6. δ-TT Induces Mitophagy in HCC Cells

As described in the Introduction section, mitophagy is a specialized form of autophagy focused on removing defective mitochondria, thus regulating mitochondrial quality control [38,39,40,41,42]. Many studies report that dysfunctional mitochondria, with marked fission, can be removed through the autophagic process [9].

We therefore investigated whether δ-TT treatment led to the removal of altered mitochondria through mitophagy. We analyzed the co-localization between mitochondria and autophagosomes through co-immunofluorescence experiments between MitoTracker and LC3. Figure 6A shows that in control cells, there was no co-localization, while in treated cells (δ-TT 25 µg/mL, 48 h), the merge between mitochondria (red) and LC3 (green) demonstrated that mitochondria were incorporated into the autophagosomes. We evaluated the expression of two proteins involved in mitophagy, Parkin and BNIP3. Parkin is a ubiquitin-ligase that accumulates on the mitochondrial outer membrane (OMM) and induces ubiquitination of mitochondrial protein, inhibition of MFN2 and activation of the clearance of depolarized and dysfunctional mitochondria through autophagy. BNIP3 is a receptor located in the OMM that recognizes LC3 and brings the mitochondrion into the autophagosome. BNIP3 is a hypoxia-inducible molecular adaptor, but it is involved in other events such as increase in ROS and mitochondrial depolarization. Furthermore, BNIP3-dependent mitophagy is preceded by mitochondrial fragmentation and translocation of Drp-1 fission protein to mitochondria [41,43].

The treatment of cells with δ-TT (25 µg/mL, 48 h) increased the expression of Parkin and BNIP3 proteins (Figure 6B). Furthermore, prolonged treatment with δ-TT (25 µg/mL, 72 h) showed a marked decrease in the expression of constitutive mitochondrial protein TOM20, demonstrating that δ-TT-induced autophagy significantly removed mitochondria (Figure 6C). TEM analysis in Figure 6D highlighted a process of mitophagy where one cell treated with δ-TT (25 µg/mL) presented an autophagosome containing a single swollen mitochondrion with a very expanded matrix space. In order to confirm that following treatment with δ-TT autophagosomes containing mitochondria were incorporated into lysosomes for their degradation, we conducted a co-immunofluorescence between MitoTracker and LysoTracker. Figure 6E shows a merged signal (yellow) in treated cells.

Furthermore, the genetic inhibition of autophagy with *ATG5 siRNA* was able to abrogate TOM20 decrease caused by treatment with δ-TT (25 µg/mL, 72 h). This result demonstrated that δ-TT-induced autophagy led to the removal of mitochondria through mitophagy (Figure 6F). Taken together, these results demonstrate that δ-TT removes dysfunctional mitochondria via mitophagy.

It is conceivable that the autophagy process removes massively dysfunctional mitochondria and determines a cellular energy deficiency that leads to cell death.

## 4. Discussion

HCC is the most common primary malignancy of the liver. Therapies approved for the treatment of advanced HCC are often ineffective due to total body toxicity and the development of acquired resistance. Therefore, it is necessary to propose new, less toxic alternative therapeutic strategies for the treatment of liver cancer.

Numerous natural products, including phytochemicals of dietary and non-dietary origin, have shown to regulate hepatocarcinogenesis and liver cancer progression [44,45,46].

In the present work we explored the role played by δ-TT, a vitamin E derivative obtained from the purification of annatto seeds, in HCC.

The main results of the present work, which was carried out in human hepatocarcinoma cell lines Huh7 and HepG2, showed that δ-TT possesses antiproliferative and cytotoxic activity. These findings are in line with previous data reporting an antitumoral action of γ-TT (another isomer of TTs) in HCC through the involvement of the JAK1, JAK2, and STAT3 signaling pathways [29], the inhibition of the Ras-Raf-MEK-ERK pathway [31], and the induction of apoptotic cell death [28].

We therefore wanted to better understand the mechanism involved in δ-TT-induced cell death in HepG2 cells. The results obtained highlight an important involvement of mitochondria in response to δ-TT treatment. Mitochondria acquired a dilated and fragmented phenotype compatible with mitochondrial fission, as confirmed by the increase in FIS-1 and DRP-1 and decrease in OPA1 and MFN-2 proteins. Among the causes favoring mitochondrial fission, there are stressful events that alter mitochondria structure and function. TEM analysis in fact confirms that the treatment with δ-TT produces “swollen” and “vesicular” dysfunctional mitochondria with mitochondrial membrane depolarization. The alteration of mitochondria structure leads to the release of mitochondrial ROS, causing cellular oxidative stress.

Our results are in agreement with numerous studies demonstrating that γ-TT and δ-TT, but also α-tocopheryl succinate (TF derivative), induces oxidative stress in different tumors (breast cancer, ovarian cancer, melanoma, and neuroblastoma) [47,48,49,50], triggering cell death. It is interesting to note that TTs exert an opposite protective antioxidant role in normal tissues (intestinal cells, vein endothelial cells, myoblasts, osteoblastic cells, muscle stem cells) [51,52,53,54,55].

Furthermore, other interesting studies highlighted that δ-TT, as well as α-TT and α-TP, exerted hepatoprotective and anti-inflammatory effects on patients with liver diseases, reducing oxidative stress involved on fibrosis, steatosis, and NAFLD [56,57,58].

In addition to this, several clinical trials have been conducted comparing the action of δ-TT and α-TF on patients with NAFLD and steatosis. Both compounds have proven effective in reducing oxidative stress, insulin resistance, steatosis, despite δ-TT being more potent. Furthermore, no adverse events occurred in any patient [59,60,61].

Autophagy is the biological process by which cells eliminate molecules or their altered organelles [62]. Indeed, autophagy analysis performed in HepG2 cells, through the expression of autophagic proteins LC3 and p62, demonstrated that δ-TT induced an active autophagic flux. Our data agree with some studies that demonstrated the ability of TTs to induce autophagy in tumors [22,24].

To date, it is known that autophagy is a dual phenomenon that can induce cell death, but also survival and resistance to cytotoxic therapies [63,64].

In this study, we explored the impact of δ-TT-induced autophagy on apoptotic cell death activation in HCC. The obtained data demonstrated that by inhibiting autophagy using 3-MA, the cytotoxic and proapoptotic activity of δ-TT was significantly reduced. Likewise, genetic autophagy inhibition, through ATG-5 silencing, counteracted the antitumoral action of δ-TT.

Mitophagy is the selected autophagy through which stressed mitochondria are chosen and acquired by autophagosomes and addressed to lysosomes for their degradation. In particular, mitochondria showing marked fission and depolarization can be removed via selective autophagy [35].

Our results demonstrated that δ-TT-induced autophagy was selective for mitochondria and that mitophagy activation led dysfunctional mitochondria within autophagolysosomes to produce a massive degradation of mitochondria. This observation is supported by the decrease in mitochondrial TOM-20 expression after δ-TT treatment and the reversion of this event by genetic inhibition of autophagy.

Many studies highlighted that mitochondria are essential for the viability of cancer cells and that the impairment of oxidative phosphorylation can reduce the survival of tumor cells [65,66]. The HCC cells are highly dependent on elevated levels of glycolysis; however, the metastatic potential of advanced HCC depends on mitochondrial respiration [67]. Indeed, an excessive mitophagy can cause the loss of functional mitochondria resulting in insufficient cellular energy and cell death. Overall, our data demonstrate that mitochondrial degradation via autophagy could represent a crucial event for the cytotoxic action of δ-TT in HepG2 cells.

## 5. Conclusions

In conclusion, we demonstrated, for the first time, that δ-TT exerts its anticancer activity through the promotion of autophagy, leading to a massive removal of dysfunctional mitochondria and inducing cell death.

The ability of δ-TT to play a hepato-protective role, and at the same time, to act as an antitumoral compound, suggests that it can represent a therapeutic opportunity for patients with advanced HCC with concomitant alterations of liver parenchyma in whom drugs with high toxicity are not recommended.

## Figures and Tables

**Figure 1 cancers-16-02654-f001:**
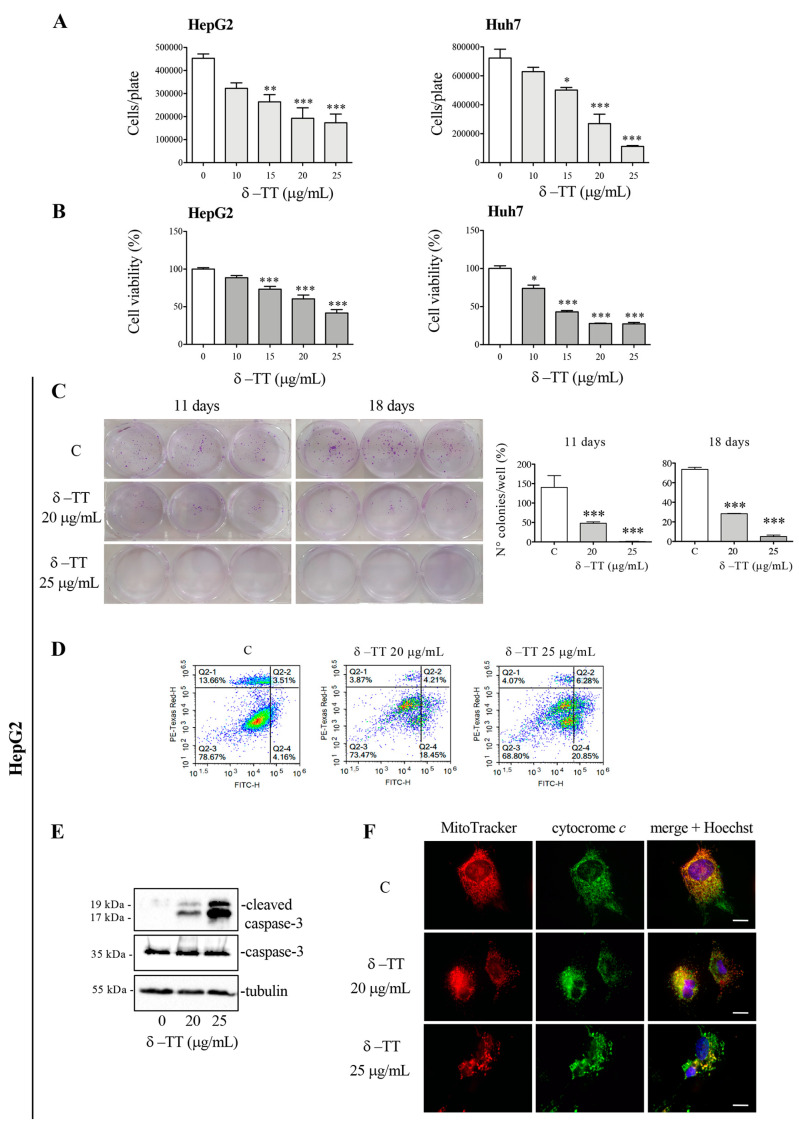
δ-TT exerts an antitumoral effect in HCC cells. (**A**) HepG2 and Huh7 cells were treated with δ-TT (10–25 μg/mL) for 48 h. Cell growth was evaluated by hemocytometer. Data represent mean values ± SEM of four independent biological samples (*n* = 4) and were analyzed by one-way analysis of variance ANOVA followed by Dunnett’s post hoc test (* *p* < 0.05 vs. C; ** *p* < 0.01 vs. C; *** *p* < 0.001 vs. C). (**B**) HepG2 and Huh7 cells were treated with δ-TT (10–25 μg/mL) for 48 h. Cell viability was then evaluated by MTT assay. Data represent mean values ± SEM of six independent biological samples (*n* = 6) and were analyzed by one-way analysis of variance ANOVA followed by Dunnett’s post hoc test (* *p* < 0.05 vs. C; *** *p* < 0.001 vs. C). (**C**) Cells were treated with δ-TT (20–25 μg/mL) for 72 h and then, after withdrawal of the treatment, were left to grow for 11 or 18 days. Colony-formation assay was performed. Statistical analysis of colonies number was analyzed by one-way analysis of variance ANOVA followed by Dunnett’s post hoc test (*** *p* < 0.001 vs. C). (**D**) Cells treated with δ-TT (20–25 μg/mL) for 72 h were stained with Annexin V/PI and analyzed by Novocyte 3000. (**E**) Cells were treated with δ-TT (20–25 μg/mL) for 72 h. WB analysis was carried out to analyze the expression levels of cleaved caspase-3 and caspase-3. Tubulin expression was evaluated as a loading control. (**F**) Cells were treated with δ-TT (20–25 μg/mL) for 48 h. At the end of treatment cells were stained with 100 nM MitoTracker Orange for 30 min to stain the mitochondria and then fixed with paraformaldehyde. The intracellular localization of cytochrome *c* was then evaluated by immunofluorescence analysis. Nuclei were stained with Hoechst. One representative of three experiments performed is shown. Images were acquired by Zeiss Axiovert 200 microscope equipped with 63×/1.4 objective lens linked to a Coolsnap Es CCD camera, scale bar 20 μm. Original western blots are presented in Appendix A.

**Figure 2 cancers-16-02654-f002:**
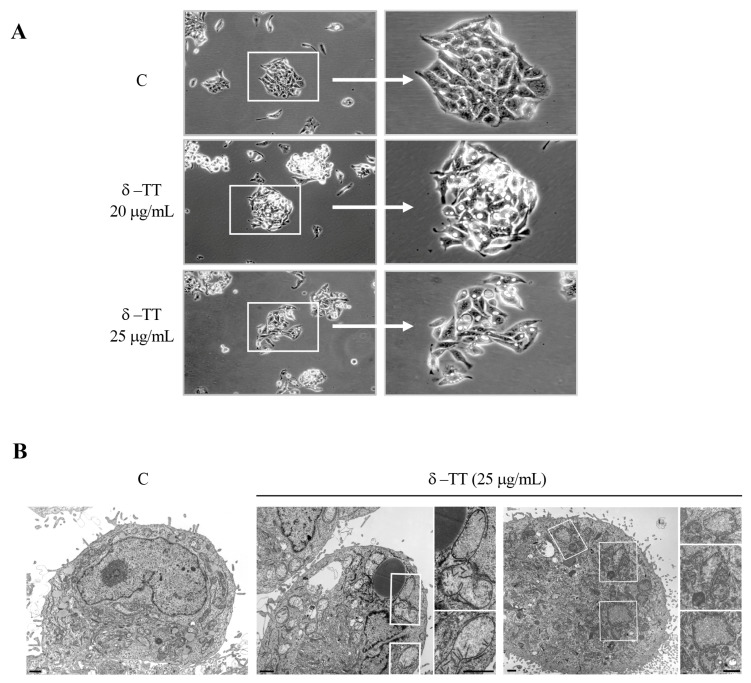
δ-TT induces cellular vacuolization and mitochondrial phenotypic alteration in HepG2. (**A**) Phase-contrast microscopy of HepG2 cells treated with δ-TT (20–25 μg/mL, 48 h). (**B**) Electron microscopy images of HepG2 cells. In control cells (C), no morphological alterations of mitochondria are evident. Cytoplasm of cells treated with δ-TT (25 μg/mL, 48 h) show both swollen mitochondria with a considerable loss of cristae (left panel) and vesicular-swollen mitochondria with a disarrangement and/or altered cristae morphology (right panel). Scale bars are 1 μm.

**Figure 3 cancers-16-02654-f003:**
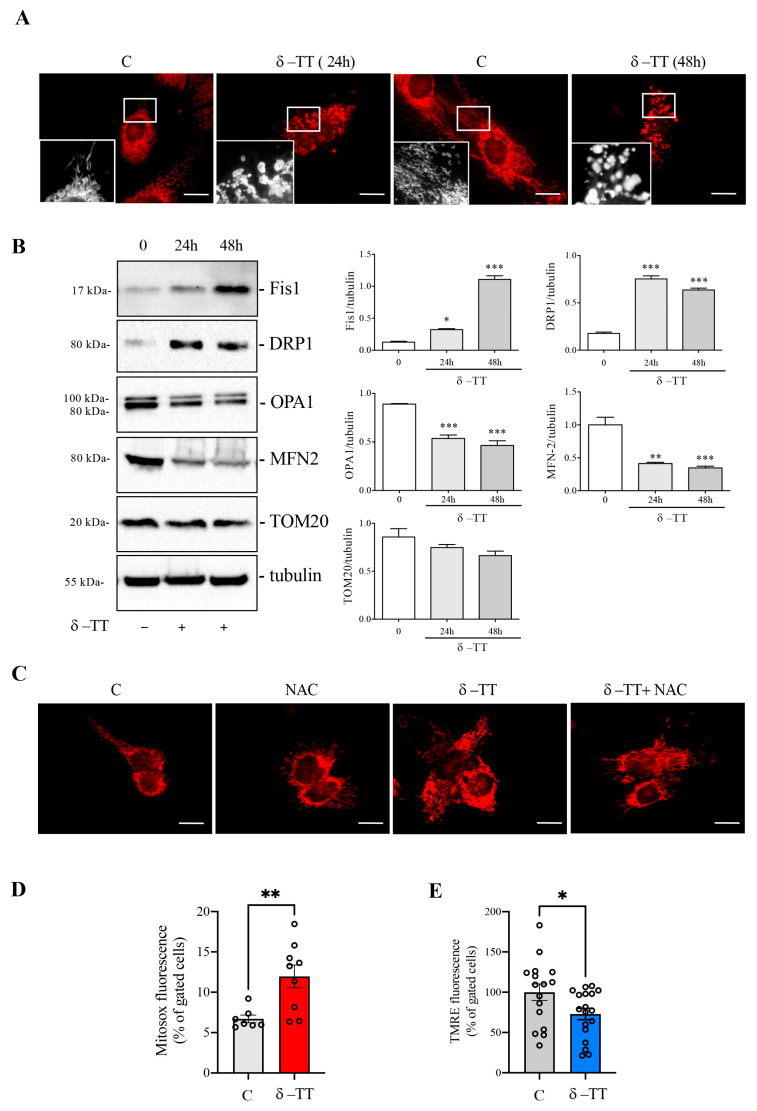
δ-TT activates mitochondrial fission and ROS release in HepG2. (**A**) Cells were treated for 24 h and 48 h with δ-TT (25 μg/mL), stained with MitoTracker Orange and then fixed with paraformaldehyde. Images were acquired by Zeiss Axiovert 200 microscope equipped with 63×/1.4 objective lens linked to a Coolsnap Es CCD camera. Scale bar 20 μm. (**B**) WB analysis was performed to investigate the expression of mitochondrial proteins Fis1, DRP1, OPA1, MNF2, and TOM20 after treatment of HepG2 cells treated with δ-TT (25 μg/mL) for 24 h and 48 h. Tubulin expression was evaluated as a loading control. Relative optical density was determined by ImageJ software. Experiments were performed independently three times and a representative blot is shown. Data represent mean values ± SEM and were analyzed by one-way analysis of variance ANOVA followed by Dunnett’s post hoc test (* *p* < 0.05 vs. C ** *p* < 0.01 vs. C; *** *p* < 0.001 vs. control). (**C**) Cells were treated with δ-TT (25 μg/mL, 24 h) in combination with or without N-acetyl-L-cysteine (NAC) (4 mM) and mitochondrial staining with MitoTracker Orange was performed. Images were acquired by Zeiss Axiovert 200 microscope equipped with 63×/1.4 objective lens linked to a Coolsnap Es CCD camera, scale bar 20 μm. (**D**) Cells were treated with δ-TT (25 μg/mL, 24 h) and stained with MitoSOX Red. MitoSOX fluorescence was analyzed by flow cytometry. Data are represented as mean values ± SEM and were analyzed by Mann–Whitney U test (** *p* < 0.01 vs. C). (**E**) Cells were treated with δ-TT (25 μg/mL, 48 h) and stained with MitoPT TMRE. Fluorescence was analyzed by flow cytometry. Data are represented as mean values ± SEM and were analyzed by Mann–Whitney U test (* *p* < 0.05 vs. C). Original western blots are presented in Appendix A.

**Figure 4 cancers-16-02654-f004:**
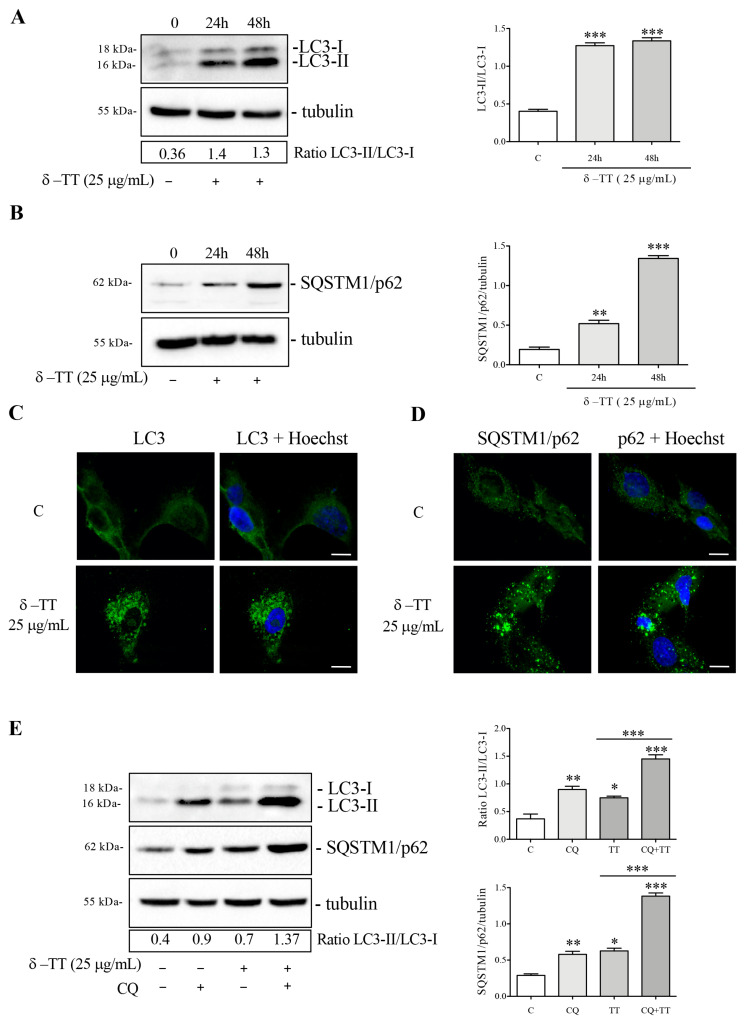
δ-TT induces autophagy in HepG2 cells. (**A**,**B**) Cells were treated with δ-TT (25 μg/mL) for 24 h and 48 h. WB analysis of LC3-II/LC3-I and SQSTM1/p62 was performed. The Tubulin expression was evaluated as a loading control. Relative optical density was determined by ImageJ software. WB was performed independently three times and a representative blot is presented. Data represent mean values ± SEM and were analyzed by one-way analysis of variance ANOVA followed by Dunnett’s post hoc test (** *p* < 0.01 vs. C; *** *p* < 0.001 vs. C). (**C**,**D**) LC3 and SQSTM1/p62 expression and intracellular localization were analyzed by immunofluorescence after treatment with δ-TT (25 μg/mL) for 48 h. Nuclei were stained with Hoechst. Images were acquired by Zeiss Axiovert 200 microscope equipped with 63×/1.4 objective lens linked to a Coolsnap Es CCD camera. Scale bar 20 μm. (**E**) Cells were treated with δ-TT (25 μg/mL) for 48 h and CQ (10 μM) for the last 24 h. LC3 and p62 levels were analyzed by WB and relative optical density were determined by ImageJ software. Experiments were performed independently three times and a representative blot is shown. Data represent mean values ± SEM and were analyzed by one-way analysis of variance ANOVA followed by Bonferroni post hoc test (* *p* < 0.05 vs. C; ** *p* < 0.01 vs. C; *** *p* < 0.001 vs. C). Original western blots are presented in Appendix A.

**Figure 5 cancers-16-02654-f005:**
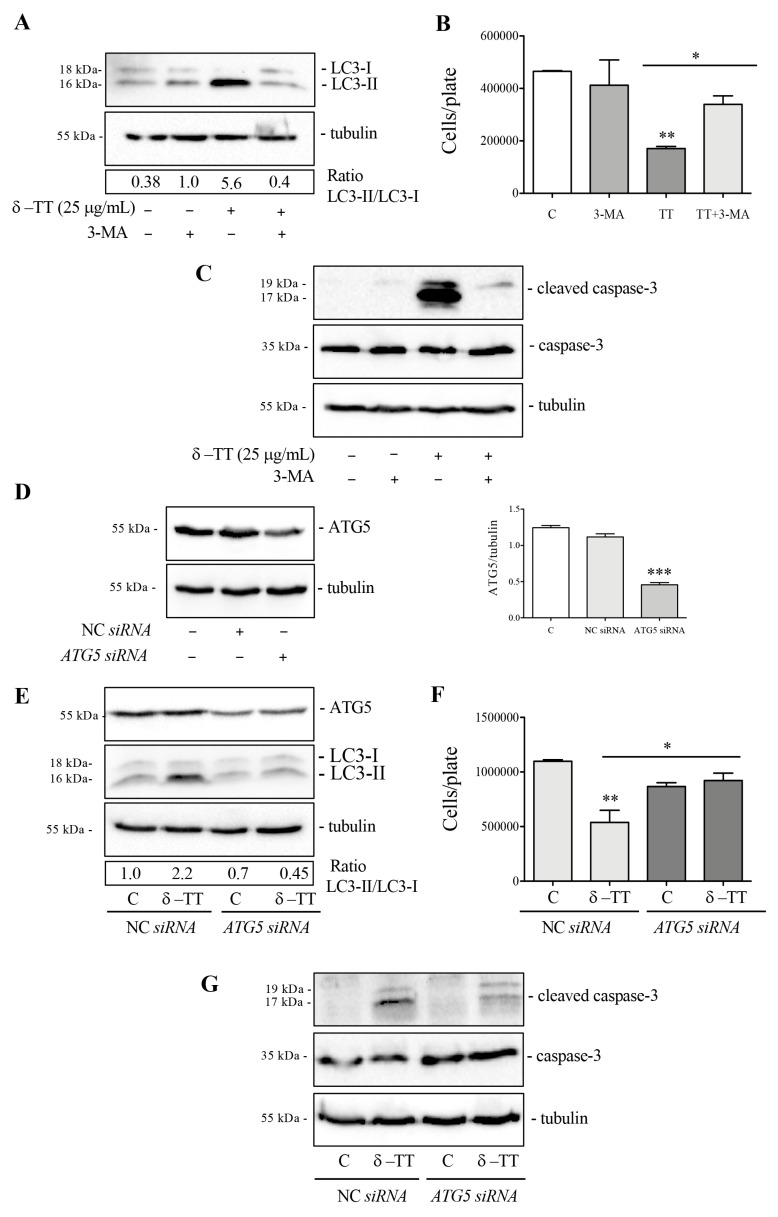
Autophagy inhibition reverts δ-TT-induced cytotoxicity in HepG2. (**A**) Cells were treated with δ-TT (25 μg/mL) and 3-MA (2 mM) for 48 h. WB was carried out to analyze the expression levels of LC3. Tubulin expression was evaluated as a loading control. (**B**) Cells were treated with δ-TT (25 μg/mL, 48 h) in the presence of autophagy inhibitor 3-MA (2 mM) and cellular proliferation was evaluated by hemocytometer. Data represent mean values ± SEM of six independent biological samples (*n* = 6). Statistical analysis was performed using one-way ANOVA followed by Bonferroni post hoc test (* *p* < 0.05; ** *p* < 0.01 vs. C). (**C**) Cells treated with δ-TT (25 μg/mL) and 3-MA (2 mM) for 72 h. The analysis of cleaved caspase-3 and caspase-3 expression was performed by WB. Tubulin expression was evaluated as a loading control. (**D**) ATG5 expression was analyzed after transfection with 50 nM negative control siRNA (NC) or *ATG5* siRNA for 72 h by WB. The Tubulin expression was evaluated as a loading control. Relative optical density was determined by ImageJ software. WB was performed independently three times and a representative blot is presented. Data represent mean values ± SEM and were analyzed by one-way analysis of variance ANOVA followed by Dunnett’s post hoc test *** *p* < 0.001 vs. C). (**E**) WB analysis of ATG5 and LC3 in cells treated with δ-TT (25 μg/mL, 48 h) was performed after transfection with NC or *ATG5* siRNA for 72 h. Tubulin expression was evaluated as a loading control. (**F**) Effect of NC *siRNA* or *ATG5* siRNA (50 nM, 72 h) on proliferation of HepG2 cells treated with δ-TT (25 μg/mL, 48 h). Data are mean ± SEM of six independent biological samples (*n* = 6). Statistical analysis was performed using one-way ANOVA followed by Bonferroni post hoc test (* *p* < 0.05; ** *p* <0.01 vs. C). (**G**) WB analysis of cleaved caspase-3 and caspase-3 in cells treated with δ-TT (25 μg/mL, 48 h) after transfection with 50 nM NC or *ATG5* siRNA for 72 h. Tubulin expression was evaluated as a loading control. Original western blots are presented in Appendix A.

**Figure 6 cancers-16-02654-f006:**
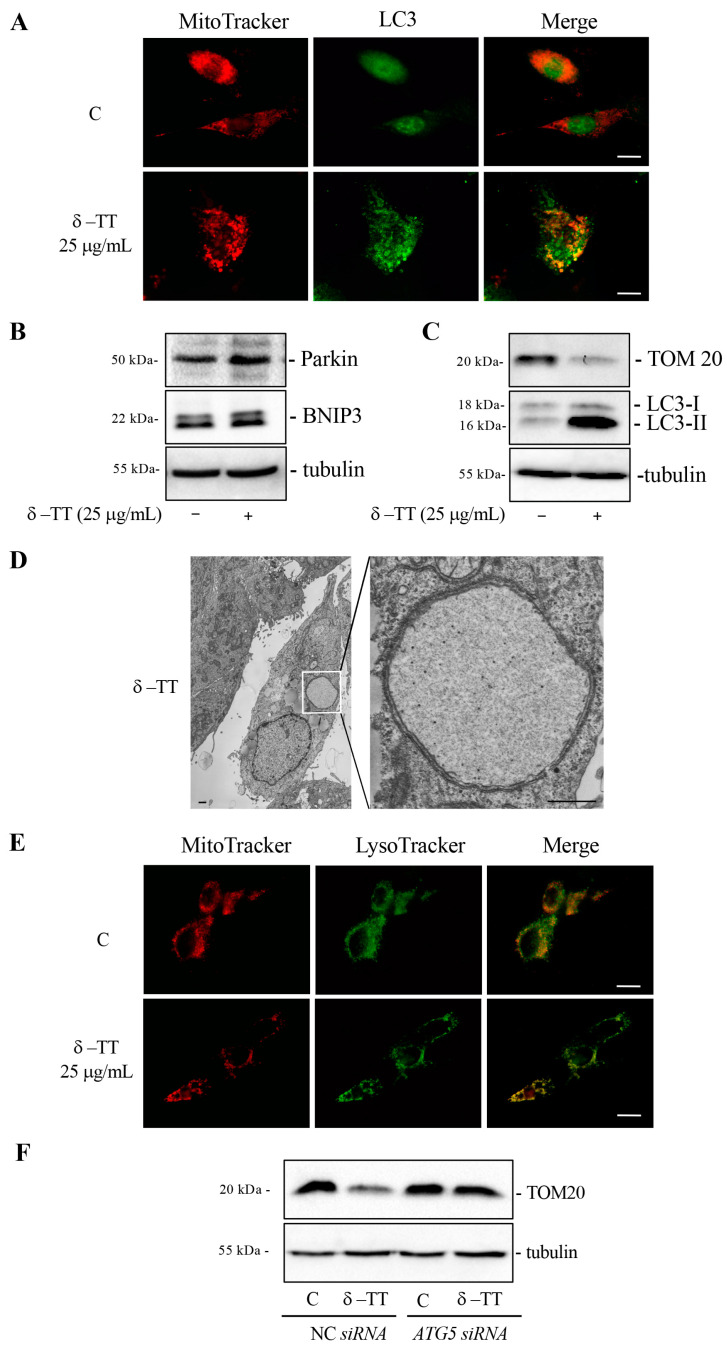
δ-TT induces mitophagy in HepG2 cells. (**A**) Cells were treated with δ-TT (25 μg/mL) for 48 h. The cells were stained with 100 nM MitoTracker Orange for 30 min to stain the mitochondria and then fixed with paraformaldehyde. Immunofluorescence for LC3 showed autophagosomes and mitochondria co-localization. Images were acquired by Zeiss Axiovert 200 microscope equipped with 63×/1.4 objective lens linked to a Coolsnap Es CCD camera, scale bar 20 μm. (**B**) WB analysis was performed to investigate the expression of Parkin and BNIP3 after treatment with δ-TT (25 μg/mL) for 48 h. Tubulin expression was evaluated as a loading control. (**C**) TOM20 and LC3 expression after treatment with δ-TT (25 μg/mL) for 72 h was analyzed by WB. Tubulin expression was evaluated as a loading control. (**D**) Electron microscopy image showing a HepG2 cell after treatment with δ-TT (25 μg/mL, 48 h). Boxed area enlarged in the right panel shows an autophagosome containing a single swollen mitochondrion with a very expanded matrix space still containing short rare cristae. Scale bars are 1 μm. (**E**) Cells were treated with δ-TT (25 μg/mL) for 48 h. The cells were stained with 100 nM MitoTracker Orange for 30 min to stain the mitochondria and 100 nM Lysotracker Green for another 45 min, then fixed with paraformaldehyde. Images were acquired by Zeiss Axiovert 200 microscope equipped with 63×/1.4 objective lens linked to a Coolsnap Es CCD camera. Scale bar is 20 μm. (**F**) WB analysis of TOM20 in cells treated with δ-TT (25 μg/mL, 72 h) after transfection with NC or *ATG5* siRNA for 72 h. Tubulin expression was evaluated as a loading control. Original western blots are presented in Appendix A.

## Data Availability

The data presented in this study are available on request from the corresponding author.

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
