# Peer review of "Anticancer Activity of Delta-Tocotrienol in Human Hepatocarcinoma: Involvement of Autophagy Induction"

_cancers, 2024, doi:10.3390/cancers16152654_

Round 1

Reviewer 1 Report

Comments and Suggestions for Authors

The manuscript entitled “Anticancer activity of Delta-Tocotrienol in human hepatocarcinoma: Involvement of autophagy induction” by Marelli et al. is an interesting study showing that delta-tocotrienol induces mitochondrial-dependent ROS generation and mitophagy. Authors claim that the induction of autophagy was related to the antiproliferative properties and apoptosis induced by Delta-Tocotrienol. However, this relationship has not fully confirmed by the study. It is critical that authors show the impact of siRNA ATG5 and Bafilomycin in the BrdU incorporation, caspase-3 and trypan blue staining in Delta-tocotrienol-treated HepG2 cells.

Author Response

Reply to Reviewer 1

We thank the Reviewer for his/her comments on our study.

With regard to your comments, we are pleased to reply as follow:

Comment. It is critical that authors show the impact of siRNA ATG5 and Bafilomycin in the BrdU incorporation, caspase-3 and Trypan blue staining in Delta-tocotrienol treated HepG2 cells.

Reply to Comment. Thanks for this comment. Based on your correct suggestions we performed caspase-3 cleaved and caspase-3 WB experiments after ATG5 siRNA in Delta-tocotrienol treated HepG2 cells. The result has been included in Figure 5 (Figure 5G).

Our cell proliferation assays were always conducted in the presence of Trypan blue. Cell count results represent only the viable cells. We apologize for not specifying this in Materials and Methods, which we have modified (lines 140-141).

Regarding further requests, unfortunately we do not have Bafilomycin and BrdU assay kit available in our laboratory therefore we are unable to conduct those experiments.

Reviewer 2 Report

Comments and Suggestions for Authors

The paper submitted by dr. M Montagnani Marelli and Coworkers deals with the role played by Vitamine E-derivative Tocotrienols during the hepatocellular carcinoma development and progression.

The Authors suggest that the activation of an autophagy process is favored by ROS mitochondrial production following δ-TT tocotrienols administration in vitro. The Authors underline a different biological role of autophagy involvement during tumor development and progression, and the contribution of two specific signal transduction pathways, the RAS/ERK and the JAK2-STAT3 pathways that are inhibited. In contrast the apoptotic cell death path is activated.

This mechanism could be highly interesting to induce during cancer pharmacological treatment.

However, some in vivo experiments should be performed at least using animal models to verify their suggestion, treating mice ± δ-TT tocotrienols and ± NAC or the two molecules together.

Some minor points should be considered:

Introduction:

1.   line 40: what does it means the word “or” between HCC and hepatoma. These two lesions are different, the two words are not synonymous.

2.   Line 73: could the Authors add a reference concerning the preventing effect of ROS during the initiation stage?

Results:

3.   Line 297: the second “caspase 3” should be 7 or which one?

4.   Line 435: What does it means “we genetically inhibited autophagy by ATG5 silencing”. The Authors inhibited specifically the expression of which gene?

Discussion:

5.   Line 518: more than “high toxicity” I should say “total body” toxicity

Comments on the Quality of English Language

 /

Author Response

Reply to Reviewer 2

We thank the Reviewer for his/her comments on our study.

With regard to your comments, we are pleased to reply as follow:

Comment. However, some in vivo experiments should be performed at least using animal models to verify their suggestion, treating mice ± δ-TT tocotrienols and ± NAC or the two molecules together.

Reply to Comment. Thanks for this suggestion. We agree that it would be very interesting to conduct further investigations on animal models. Now, we do not have the possibility to conduct this study as the authorizations for the use of animals require very long times. In the future we would like to conduct the suggested experiment.

Minor point:

  1.   Line 40: what does it means the word “or” between HCC and hepatoma. These two lesions are different, the two words are not synonymous. Reply: We have eliminated “hepatoma” 
  2.  Line 73: could the Authors add a reference concerning the preventing effect of ROS during the initiation stage? Reply: We have added two references (Ref.10 and Ref.11- line 74). 
  3.  Line 297: the second “caspase 3” should be 7 or which one? Reply: The first “cleaved caspase-3” is the active form of caspase-3; the second “caspase-3” is the inactive procaspase- 3. 
  4.  Line 435: What does it means “we genetically inhibited autophagy by ATG5 silencing”. The Authors inhibited specifically the expression of which gene? Reply: The silenced gene is ATG5; we have modified the sentence to make the concept clearer (line 438). 
  5.  Line 518: more than “high toxicity” I should say “total body” toxicity. Reply: We replaced high toxicity with total body toxicity (line 523).   

Round 2

Reviewer 1 Report

Comments and Suggestions for Authors

The manuscript entitled “Anticancer activity of Delta-Tocotrienol in human hepatocarcinoma: Involvement of autophagy induction” by Marelli et al. is an interesting study showing that delta-tocotrienol induces mitochondrial-dependent ROS generation and mitophagy. Authors claim that the induction of autophagy was related to the antiproliferative properties and apoptosis induced by Delta-Tocotrienol. Regarding the previous version, the authors have addressed one of the queries regarding the effect of siRNA ATG5 on caspase-3 activation in HepG2 cells.